# Healthy lifestyle and life expectancy in people with multimorbidity in the UK Biobank: A longitudinal cohort study

Yogini V. Chudasama[1,2]*, Kamlesh Khunti[1,2], Clare L. Gillies[1], Nafeesa N. Dhalwani[1], Melanie J. Davies[1,3], Thomas Yates[1,3], Francesco Zaccardi[1]

1 Diabetes Research Centre, Leicester General Hospital, University of Leicester, Leicester, United Kingdom,
2 National Institute for Health Research (NIHR) Applied Research Collaboration—East Midlands (ARC-EM) Leicester Diabetes Centre, Leicester, United Kingdom, 3 National Institute for Health Research (NIHR) Leicester Biomedical Research Centre, Leicester Diabetes Centre, Leicester, United Kingdom

* yc244@leicester.ac.uk

## Abstract

### Background

Whether a healthy lifestyle impacts longevity in the presence of multimorbidity is unclear. We investigated the associations between healthy lifestyle and life expectancy in people with and without multimorbidity.

### Methods and findings

A total of 480,940 middle-aged adults (median age of 58 years [range 38–73], 46% male, 95% white) were analysed in the UK Biobank; this longitudinal study collected data between 2006 and 2010, and participants were followed up until 2016. We extracted 36 chronic conditions and defined multimorbidity as 2 or more conditions. Four lifestyle factors, based on national guidelines, were used: leisure-time physical activity, smoking, diet, and alcohol consumption. A combined weighted score was developed and grouped participants into 4 categories: very unhealthy, unhealthy, healthy, and very healthy. Survival models were applied to predict life expectancy, adjusting for ethnicity, working status, deprivation, body mass index, and sedentary time. A total of 93,746 (19.5%) participants had multimorbidity. During a mean follow-up of 7 (range 2–9) years, 11,006 deaths occurred. At 45 years, in men with multimorbidity an unhealthy score was associated with a gain of 1.5 (95% confidence interval [CI] −0.3 to 3.3; P = 0.102) additional life years compared to very unhealthy score, though the association was not significant, whilst a healthy score was significantly associated with a gain of 4.5 (3.3 to 5.7; P < 0.001) life years and a very healthy score with 6.3 (5.0 to 7.7; P < 0.001) years. Corresponding estimates in women were 3.5 (95% CI 0.7 to 6.3; P = 0.016), 6.4 (4.8 to 7.9; P < 0.001), and 7.6 (6.0 to 9.2; P < 0.001) years. Results were consistent in those without multimorbidity and in several sensitivity analyses. For individual lifestyle factors, no current smoking was associated with the largest survival benefit. The main limitations were that we could not explore the consistency of our results using a more restrictive definition of multimorbidity including only cardiometabolic conditions, and participants were not representative of the UK as a whole.

**Data Availability Statement:** The data that support the findings of this study are available from the UK Biobank project site, subject to registration and

application process. Further details can be found at https://www.ukbiobank.ac.uk.

**Funding:** YC is funded by a University of Leicester College of Medicine, Biological Sciences and Psychology PhD studentship in collaboration with Collaboration for Leadership in Applied Health Research and Care East Midlands (CLAHRC EM), now recommissioned as NIHR Applied Research Collaboration East Midlands (ARC EM). The funders had no role in study design, data collection and analysis, decision to publish, or preparation of the manuscript.

**Competing interests:** I have read the journal's policy and the authors of this manuscript have the following competing interests. FZ is funded with an unrestricted educational grant from the NIHR CLAHRC East Midlands to the University of Leicester. KK has acted as a consultant and speaker for Novartis, Novo Nordisk, Sanofi-Aventis, Lilly, Servier, and Merck Sharp & Dohme. He has received grants in support of investigator and investigator-initiated trials from Novartis, Novo Nordisk, Sanofi-Aventis, Lilly, Pfizer, Boehringer Ingelheim, and Merck Sharp & Dohme. KK has received funds for research and honoraria for speaking at meetings and has served on advisory boards for Lilly, Sanofi-Aventis, Merck Sharp & Dohme, and Novo Nordisk. MJD has acted as consultant, advisory board member, and speaker for Novo Nordisk, Sanofi-Aventis, Lilly, Merck Sharp & Dohme, Boehringer Ingelheim, AstraZeneca, and Janssen; an advisory board member for Servier and Gilead Sciences Ltd; and as a speaker for NAPP, Mitsubishi Tanabe Pharma Corporation, and Takeda Pharmaceuticals International Inc. TY has received funding from the Leicester NIHR Leicester BRC. All other authors have declared that no competing interests exist.

**Abbreviations:** CI, confidence interval; HR, hazard ratio; MET, metabolic equivalent of task; NHS, National Health Service; NICE, National Institute for Health and Care Excellence; ONS, Office for National Statistics; QoF, Quality and Outcomes Framework; STROBE, Strengthening the Reporting of Observational Studies in Epidemiology.

## Conclusions

In this analysis of data from the UK Biobank, we found that regardless of the presence of multimorbidity, engaging in a healthier lifestyle was associated with up to 6.3 years longer life for men and 7.6 years for women; however, not all lifestyle risk factors equally correlated with life expectancy, with smoking being significantly worse than others.

---

### Author summary

#### Why was this study done?

- People with multimorbidity (presence of 2 or more chronic conditions) have poorer health outcomes and a higher mortality risk compared to people without multimorbidity.

- A healthy lifestyle has been associated with a longer life expectancy. To our knowledge, no study to date has investigated this relationship in relation to the presence of multimorbidity.

- Most studies used a combined score that did not account for the differential impact of each lifestyle factor on the risk of death.

#### What did the researchers do and find?

- We investigated the association between healthy lifestyle and individual risk factors with life expectancy in relation to the presence of multimorbidity.

- We found that an overall healthy lifestyle largely counterbalances the negative association between multimorbidity and life expectancy.

#### What do these findings mean?

- A healthier lifestyle is consistently associated with a longer life expectancy across various individual risks and irrespective of the presence of multiple long-term medical conditions.

- Public health recommendations about a healthy lifestyle to reduce the risk of developing chronic long-term conditions equally apply to individuals who have already multimorbidity.

## Introduction

Multimorbidity, commonly defined as the presence of 2 or more long-term physical or mental health conditions [1,2], has recently become a major worldwide epidemic [3]. Considerable evidence exists on the increased prevalence and the negative impact that multimorbidity has on patients, family, carers, and healthcare systems [1]. Nevertheless, there is still limited

research on approaches to self-managing multimorbidity [3,4]. People who engage in a healthy lifestyle, such as eating a balanced diet, exercising regularly, and avoiding smoking and excess alcohol consumption, have many health benefits, especially in terms of improved longevity [5–7]; in particular, a lower alcohol intake and greater levels of physical activity have been associated with proportionally larger effects on life expectancy in large observational studies [8,9]. However, whether and to what extent a healthy lifestyle impacts on longevity in people with multimorbidity is less clear. Clarifying this uncertainty may have important individual, clinical, and public health implications, in view of the rapidly increasing trends in the prevalence of multimorbidity [3].

Life expectancy estimates are easier to understand for both the public and healthcare professionals and have become a common metric for establishing public health priorities. To date, no study has explored the association of both individual and combined lifestyle factors such as smoking, diet, and alcohol intake with life expectancy, in relation to the presence of multimorbidity [3]. Only one study assessed the relationship of combined healthy lifestyle with life expectancy in people with one or more chronic conditions [10], while the remaining investigations included individuals from the general population, where the findings showed that a combined healthy lifestyle was associated with a life expectancy between 5.4 and 18.9 years longer compared to the unhealthiest group (S1 Table) [5–7,10–19]. Most of these studies used a combined score that did not account for the differential impact of each lifestyle factor on the risk of death while the magnitude of the association may vary across multiple lifestyle factors [7,20,21].

To clarify this uncertainty, we have investigated in a contemporary population the association between individual risk factors and a healthy lifestyle with life expectancy in relation to the presence of multimorbidity.

## Methods

This study is reported as per the Strengthening the Reporting of Observational Studies in Epidemiology (STROBE) guideline (S1 Checklist) following a pre-specified protocol [22]; local Institutional Review Board ethics approval was not necessary for this study.

### Study population

We used data from the UK Biobank study (Application Number 14146). UK Biobank included 502,629 middle-aged (38–73 years) adults recruited from 22 sites across England, Wales, and Scotland with baseline measures collected between 2006 and 2010 and with data linked to mortality records [22]. Written informed consent was obtained prior to data collection; UK Biobank was approved by the National Health Service (NHS) National Research Ethics Service (16/NW/0274; ethics approval for UK Biobank studies) [23]. To minimise reverse causality, we excluded participants who died within the first 2 years of follow-up ($n$ = 2,516) [24]. Participants who withdrew from the study ($n$ = 91), whose age during follow-up was less than 45 years ($n$ = 30), who had missing lifestyle data ($n$ = 16,503), or who had missing covariate data ($n$ = 2,549) were excluded from the analysis (S1 Fig).

### Multimorbidity

UK Biobank collected self-reported medical information based on physician diagnosis. To define multimorbidity, 3 sources were used to select long-term cardiovascular, non-cardiovascular, or mental health conditions. The first included conditions from the Quality and Outcomes Framework (QoF), which reports the most common diseases in the UK [25]; the second is a large UK-based study, containing 40 of the recommended core disorders for any

multimorbidity measure [1]; and the last is a systematic review on multimorbidity indices that included 17 conditions [26]. Based on these sources and the data available in UK Biobank, we selected a total of 36 chronic conditions: participants with 2 or more of these conditions were classified as having multimorbidity (S1 Text). Some of the diseases previously considered in the definitions of multimorbidity have not been included in this analysis as they have been used in the statistical modelling (i.e., obesity and alcohol, as body mass index is a model covariate and alcohol consumption is part of the lifestyle score); conversely, others (anaemia, meningitis, tuberculosis, and vestibular disorders) have been added as they were deemed clinically relevant. The combination of these 3 sources to identify the conditions has been adopted also in previous studies and [8,27], particularly by including the most common QoF diseases, enhances the generalisability of the results.

## Mortality

Mortality data were obtained from the NHS Information Centre for participants from England and Wales and the NHS Central Register for participants from Scotland. Data for survivors were censored on 31 January 2016 for England and Wales and 30 November 2015 for Scotland.

## Healthy lifestyle

Four well-known healthy lifestyle factors, based on national guidelines [28–32], were used in this study: leisure-time physical activity, smoking, diet (fruit and vegetables), and alcohol consumption; information on these factors was collected from an in-person baseline interview at the UK Biobank centre (http://biobank.ctsu.ox.ac.uk/crystal/search.cgi).

   For leisure-time physical activity, participants were asked "In the last four weeks, did you spend any time doing the following: walking for pleasure, light DIY (do-it-yourself, i.e., home maintenance and improvement and gardening activities), heavy DIY (e.g., using heavy tools, weeding, lawn mowing, digging, carpentry), strenuous sports (i.e., sports that make you sweat or breathe hard), other exercises (e.g., swimming, cycling, keep fit, bowling); none of the above." Participants could select more than one activity and were asked to quantify their participation by frequency (i.e., number of times in the previous 4 weeks) and duration. The intensity was expressed in terms of standardised metabolic equivalent of task (MET) values: 3.5 METs for walking for pleasure; 5.5 METs for heavy DIY; 8.0 METs for strenuous sports; 4.0 METs for other activities [33]. We did not include light DIY within our definition, since we were specifically investigating moderate to vigorous intensity physical activities. The total weekly leisure-time physical activity (MET-minutes/week) was calculated by multiplying the frequency, duration, and the MET values [33]. Regular physical activity was defined as meeting the current global health recommendations for physical activity (150 minutes of moderate activity or 75 minutes of vigorous activity or an equivalent combination) [28,32], which equated to ≥500 MET-minutes/week, or no regular physical activity (<500 MET-minutes/week). Smoking was categorised as not current smoker or current smoker at the time of assessment. A healthy diet was based on eating at least 5 portions of a variety of fruit and vegetables every day following the NHS guideline [29]. To calculate the portions, we used combined responses for fresh fruit (pieces), dried fruit (pieces), salad/raw vegetable (heaped tablespoons), and cooked vegetable (heaped tablespoons): these portions were grouped as ≥5 portions/day (meet fruit/vegetable guidelines) or <5 portions/day (do not meet fruit/vegetable guidelines). The UK Biobank asked participants for the number of pints of beer, glasses of wine, and measures of spirit consumed in the last week. Alcoholic drinks differ in the amount of alcohol content, therefore each drink was converted into equivalent standard units, where 1 unit contains

10 ml of ethyl alcohol [34]. The guidelines from the Office for National Statistics (ONS) were used as they report the most updated method of converting volumes to units [34]. Total weekly units of alcohol were calculated by adding the units of beer, wine, and spirits, and participants were grouped as reporting none/moderate alcohol consumption (0–14 units per week) or excess alcohol consumption (>14 units per week), based on the NHS guidelines [30].

A weighted healthy lifestyle score, combining the 4 risk factors, was computed (details are reported in the statistical analysis).

## Confounders

All models were adjusted for ethnicity (white or non-white), socioeconomic status (measured using the Townsend deprivation index, which combines census data on housing, employment, and social class based on the postal code of participants), employment status (working, retired, or other [unemployed, looking after home and/or family, unable to work because of sickness or disability, unpaid/voluntary work, full/part time student, or did not answer]), body mass index calculated during the physical assessments, and total sedentary time estimated from the sum of self-reported hours spent watching television, using the computer, and driving during a typical day: values greater than 24 hours per day were excluded, and in those reporting over 16 hours sedentary time values were winsorized at 16 hours.

## Statistical analysis

To account for potential differences in the association between each lifestyle factor and mortality risk, a weighted healthy lifestyle score was computed: β coefficients of each healthy lifestyle factor were estimated using a flexible parametric Royston-Parmar proportion-hazards model that included all 4 lifestyle factors and death as an outcome [35]. Participants were classified into 2 groups: 0 (no regular physical activity; current smoking; <5 portions/day of fruit/vegetable; excess alcohol intake) and 1 (regular physical activity; no current smoking; ≥5 portions/day of fruit/vegetable; none/moderate alcohol consumption). The binary lifestyle variables for each participant were then multiplied by the standardised weighted β coefficients, summed, and grouped in 4 ordered categories (further details are provided in S2 Text): very unhealthy (score 0–0.25; reference group), unhealthy (≥0.25–0.50), healthy (≥0.50–0.75), and very healthy (≥0.75–1).

Separate models were fitted for those with and those without multimorbidity, and for men and women. Hazard ratios (HRs) and corresponding 95% confidence intervals (CIs) of all-cause mortality were calculated in complete-case analysis with age as time scale: estimates were firstly obtained for the lifestyle categories and then for each individual lifestyle factor. The calculation of years of life lost (i.e., difference in average life expectancy) involved a two-step process. First, residual life expectancy was estimated as the area under the survival curve up to 100 years old, conditional on surviving at ages 45 to 100 years old (1-year intervals); survival curves were predicted for each individual and averaged over individuals. Second, years of life lost and 95% CIs were calculated as the difference between the areas under 2 survival curves, between lifestyle categories and for each individual lifestyle factors. All analyses were adjusted for confounders (i.e., ethnicity, employment status and continuous effect for deprivation, BMI, and sedentary time).

We conducted 4 sensitivity analyses to assess the robustness of our results (S2 Table). In the first, we derived β coefficients using a random one-third of the dataset and estimated the weighted score in the remaining two-thirds (S2 Text); we also analysed data after imputing missing covariates (S3 Text). In a second sensitivity analysis, we re-performed all calculations using a continuous score obtained from continuous lifestyle variables (further details are

reported in S4 Text). In the third, we used a more homogenous definition of multimorbidity, limited to cardiometabolic conditions (diabetes and cardiovascular diseases: stroke, myocardial infarction, heart failure, angina or peripheral vascular disease). In the fourth, we complemented our main results with analyses using a total score derived from the sum of each score, which therefore ranged from 0 (none of the "healthy" lifestyle factors present) to 4 (all present). Lastly, we estimated HRs and years of life lost in participants without multimorbidity who were matched to those with multimorbidity (further details are provided in S5 Text).

Stata version 16.0 was used to manipulate data and perform the survival analyses (stpm2 command) [35]. Results are reported with two-sided 95% CI.

## Results

### Baseline characteristics

In 480,940 participants, the 5 most prevalent chronic conditions for men were hypertension (29.6%), asthma (10.7%), cancer (6.3%), diabetes (5.8%), and angina (4.6%) and for women hypertension (22.7%), asthma (12.3%), cancer (9.8%), depression (6.7%), and migraine (4.2%); a total of 93,746 (19.5%) participants had multimorbidity (S3 and S4 Tables). Most participants were white (94.8%), with a median (range) age of 58 (38–73) years. Compared to participants without multimorbidity, those with multimorbidity were older (61 [54–65]) versus 57 [49–63] years, respectively) and more likely to live in deprived areas, be retired (45.4% versus 30.2%, respectively), and spend more time in sedentary activities (Table 1).

The lifestyle factors at baseline showed fewer participants with multimorbidity engaging in regular physical activity compared to those without multimorbidity (44.6% versus 54.0%, respectively) but slightly more reported a healthy diet (39.2% versus 37.7%) and consumed none or a moderate amount of alcohol (66.5% versus 61.5%). There was a similar proportion of participants who were not currently smokers (89.2% versus 89.8%). For the combined healthy lifestyle score, in participants with multimorbidity 8.3% were very unhealthy, 2.4% unhealthy, 34.8% healthy, and 54.4% very healthy; corresponding estimates in participants without multimorbidity were 7.4%, 2.8%, 30.6%, and 59.2% (Table 1).

### Healthy lifestyle

During a mean follow-up of 7 (range, 2–9) years and 3.34 million person-years, 11,006 deaths were recorded. Compared to the reference group (very unhealthy), the adjusted HRs of mortality were lower in healthier groups in both men and women, ranging from HR 0.83 (95% CI 0.66–1.03; $P = 0.096$) to 0.40 (0.34–0.47; $P < 0.001$) in those with multimorbidity and from 0.84 (0.68–1.04; $P = 0.111$) to 0.35 (0.32–0.39; $P < 0.001$) in those without (Fig 1).

Life expectancy rose as the level of healthy lifestyle increased (Table 2 and Fig 2). After covariate adjustments, at the age of 45 years in men with multimorbidity, an unhealthy score was associated with a gain of 1.5 (95% CI −0.3 to 3.3; $P = 0.102$) additional life years compared to very unhealthy; a healthy score with 4.5 (3.3–5.7; $P < 0.001$) years; and a very healthy score with 6.3 (5.0–7.7; $P < 0.001$) years. Corresponding estimates in women with multimorbidity were 3.5 (95% CI 0.7–6.3; $P = 0.016$), 6.4 (4.8–7.9; $P < 0.001$), and 7.6 (6.0–9.2; $P < 0.001$) years. In men without multimorbidity, an unhealthy score was associated with a gain of 2.8 (95% CI 1.5–4.1; $P < 0.001$) additional life years compared to very unhealthy, a healthy score with 5.7 (4.7–6.7; $P < 0.001$), and a very healthy score with 7.6 (6.5–8.6; $P < 0.001$) years. Corresponding estimates in women were 1.3 (95% CI −0.3 to 3.0; $P = 0.111$), 6.0 (4.9–7.2; $P < 0.001$), and 6.5 (5.4–7.6; $P < 0.001$) years. The pattern of results was similar at the age of 65 years (Table 2 and Fig 2).

**Table 1. Baseline characteristics of participants by multimorbidity status.**

| Characteristics | With multimorbidity (*n* = 93,746) | Without multimorbidity (*n* = 387,194) |
|---|---|---|
| **Age, median [IQR], y** | 61 [54–65] | 57 [49–63] |
| **Sex** | | |
| Women | 50,298 (53.7) | 211,814 (54.7) |
| Men | 43,448 (46.4) | 175,380 (45.3) |
| **Ethnicity** | | |
| White | 88,863 (94.8) | 367,234 (94.8) |
| Non-white | 4,883 (5.2) | 19,960 (5.2) |
| **Employment status** | | |
| Working | 34,438 (41.0) | 239,910 (62.0) |
| Retired | 42,538 (45.4) | 116,820 (30.2) |
| Other[a] | 12,770 (13.6) | 30,464 (7.8) |
| **Deprivation index,[b] mean [SD]** | −0.9 [3.3] | −1.4 [3.0] |
| **BMI, mean [SD], kg/m²** | 29.0 [5.5] | 27.0 [4.5] |
| **Sedentary behaviour,[c] mean [SD], h** | 5.4 [2.5] | 5.0 [2.3] |
| **Lifestyle factors[d]** | | |
| Regular physical activity | 41,809 (44.6) | 209,186 (54.0) |
| Not currently smoking | 83,633 (89.2) | 347,499 (89.8) |
| Healthy diet | 36,734 (39.2) | 145,946 (37.7) |
| None/moderate alcohol consumption | 62,322 (66.5) | 238,194 (61.5) |
| **Healthy lifestyle categories** | | |
| Very unhealthy | 7,822 (8.3) | 28,814 (7.4) |
| Unhealthy | 2,291 (2.4) | 10,881 (2.8) |
| Healthy | 32,624 (34.8) | 118,317 (30.6) |
| Very healthy | 51,009 (54.4) | 229,182 (59.2) |

Shown are numbers (%) unless stated otherwise.

[a]Other = unemployed, student, volunteer, or missing.

[b]Deprivation = Townsend deprivation index was used as a measure of socioeconomic status, which combines census data on housing, employment, social class, and car availability based on the postal code of participants.

[c]Sedentary = total number of self-reported hours spent watching television, using the computer, or driving.

[d]Regular physical activity: ≥500 MET-minutes/week; None/moderate alcohol consumption: 0 to 14 units of alcohol a week; Healthy diet: at least 5 portions of fruit and vegetables every day.

**Abbreviations:** BMI, body mass index; IQR, interquartile range; MET, metabolic equivalent of task

### Individual lifestyle factor

The associations between individual healthy lifestyle factors and survival are presented in Table 3. The largest survival difference was observed for the risk factor smoking: the adjusted mortality rate comparing non-current versus current smoker in participants with multimorbidity was 46% lower (HR 0.54 [95% CI 0.49–0.60; $P < 0.001$]) in men and 52% lower (HR 0.48 [0.42–0.55; $P < 0.001$]) in women; corresponding estimates in participants without multimorbidity were 0.45 (0.41–0.48; $P < 0.001$) and 0.44 (0.40–0.49; $P < 0.001$). At the age of 45 years, in participants with multimorbidity who do not currently smoke, the estimated life expectancy compared to those who smoke was 4.9 (95% CI 3.8–6.1; $P < 0.001$) years longer in men and 5.9 (4.6–7.3; $P < 0.001$) years longer in women; in those without multimorbidity, corresponding estimates were 5.9 (5.0–6.8; $P < 0.001$) and 5.8 (4.8–6.7; $P < 0.001$) years.

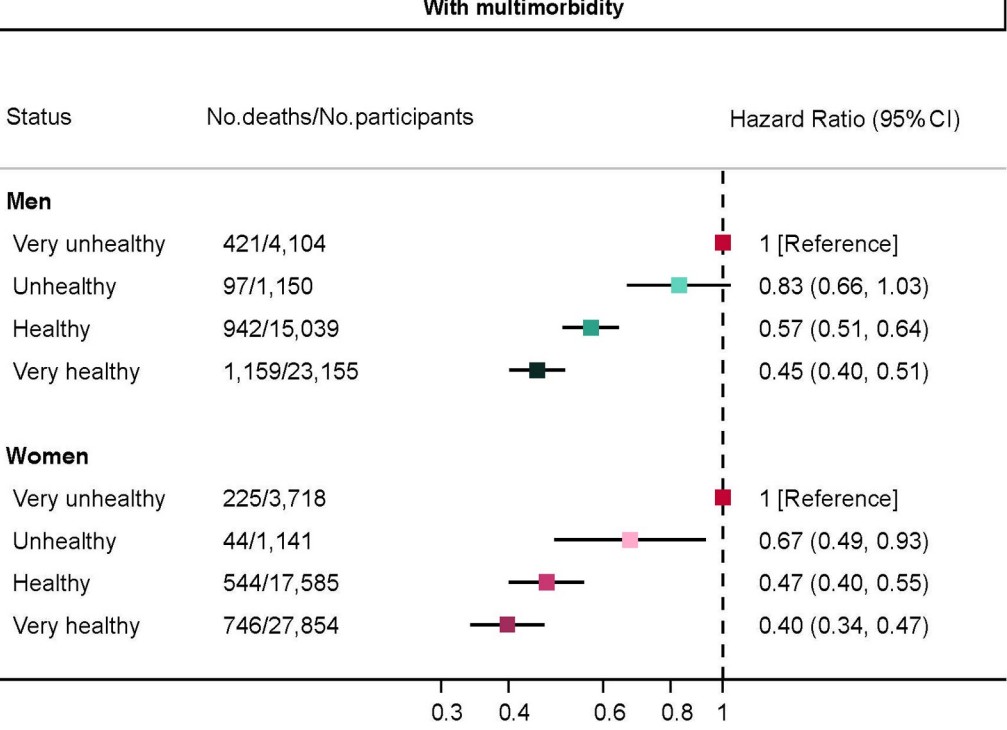

**Fig 1. HRs of death by lifestyle score.** Models adjusted for ethnicity (white, non-white), working status (working, retired, other), deprivation (continuous), body mass index (continuous), sedentary time (continuous). CI, confidence interval; HR, hazard ratio; No., number.

**Table 2. Years of life gained at age 45 and 65 years by lifestyle score.**

| Healthy lifestyle category | With multimorbidity | | | | Without multimorbidity | | | |
|---|---|---|---|---|---|---|---|---|
| | Men (*n* = 43,448) | *P* value | Women (*n* = 50,298) | *P* value | Men (*n* = 175,380) | *P* value | Women (*n* = 211,814) | *P* value |
| **Years of life gained [95% CI], 45 y** | | | | | | | | |
| Very unhealthy | Reference | - | Reference | - | Reference | - | Reference | - |
| Unhealthy | 1.50 [−0.30 to 3.30] | 0.102 | 3.48 [0.65 to 6.31] | 0.016 | 2.77 [1.49 to 4.05] | <0.001 | 1.34 [−0.31 to 2.99] | 0.111 |
| Healthy | 4.52 [3.30 to 5.73] | <0.001 | 6.36 [4.79 to 7.94] | <0.001 | 5.66 [4.65 to 6.66] | <0.001 | 6.03 [4.90 to 7.15] | <0.001 |
| Very healthy | 6.33 [4.98 to 7.69] | <0.001 | 7.59 [6.01 to 9.16] | <0.001 | 7.56 [6.47 to 8.64] | <0.001 | 6.49 [5.39 to 7.59] | <0.001 |
| **Years of life gained [95% CI], 65 y** | | | | | | | | |
| Very unhealthy | Reference | - | Reference | - | Reference | - | Reference | - |
| Unhealthy | 1.20 [−0.26 to 2.65] | 0.106 | 2.94 [0.53 to 5.36] | 0.017 | 2.39 [1.26 to 3.52] | <0.001 | 1.19 [−0.28 to 2.66] | 0.112 |
| Healthy | 3.70 [2.67 to 4.73] | <0.001 | 5.43 [4.07 to 6.79] | <0.001 | 4.96 [4.05 to 5.88] | <0.001 | 5.42 [4.40 to 6.45] | <0.001 |
| Very healthy | 5.26 [4.09 to 6.42] | <0.001 | 6.50 [5.13 to 7.86] | <0.001 | 6.70 [5.69 to 7.70] | <0.001 | 5.85 [4.84 to 6.85] | <0.001 |

Models adjusted for ethnicity (white, non-white), working status (working, retired, other), deprivation (continuous), body mass index (continuous), sedentary time (continuous).

**Abbreviation:** CI, confidence interval

Regular physical activity was associated with the second highest survival benefit. At the age of 45 years, regular physical activity was associated with 2.5 (95% CI 1.8–3.2; *P* < 0.001) years longer life expectancy in men and 1.9 (1.1–2.7; *P* < 0.001) in women with multimorbidity; in those without multimorbidity, corresponding estimates were 1.8 (1.3–2.3; *P* < 0.001) and 0.9 (0.4–1.4; *P* < 0.001) years. The years of life gained were smaller for alcohol consumption and healthy diet.

## Sensitivity analyses

The main results were confirmed in the first sensitivity analysis, using a third of the population to estimate the weighted score (S5 Table) or following imputation of missing data (S6 and S7 Tables). In the second sensitivity analysis, the pattern of the main results showing a similar benefit regardless of the presence of multimorbidity was confirmed when a continuous score was obtained from the entire population (S2 Fig and S8 Table), in one-third of the population (S3 Fig and S9 Table), or following imputation of missing data (S4 Fig and S10 Table). When the outcome was limited to cardiometabolic multimorbidity (third sensitivity analysis), the number of participants and events was significantly lower compared to multimorbidity defined using the main definition (3,804 versus 93,746 individuals), particularly women: this resulted in imprecise estimates of HR and years of life gained across groups defined by the weighted score (S11 Table). Similarly, very few participants and events were observed when investigating each lifestyle factor, yet the pattern was qualitatively similar to the main results indicating a greater relevance on life expectancy of physical activity and smoking compared to alcohol consumption and healthy diet (S12 Table). Although years of life gained were slightly greater comparing heathiest versus unhealthiest groups, the main results were largely confirmed using a score obtained from the sum of each "healthy" lifestyle and multimorbidity as outcome (S13 Table); however, imprecise or no estimates were obtained using the same score and cardiometabolic multimorbidity as outcome, due to very few participants and events (S14 Table). Lastly, the main results were confirmed in the cohort of participants without multimorbidity matched to those with multimorbidity (S5 Fig and S15 Table).

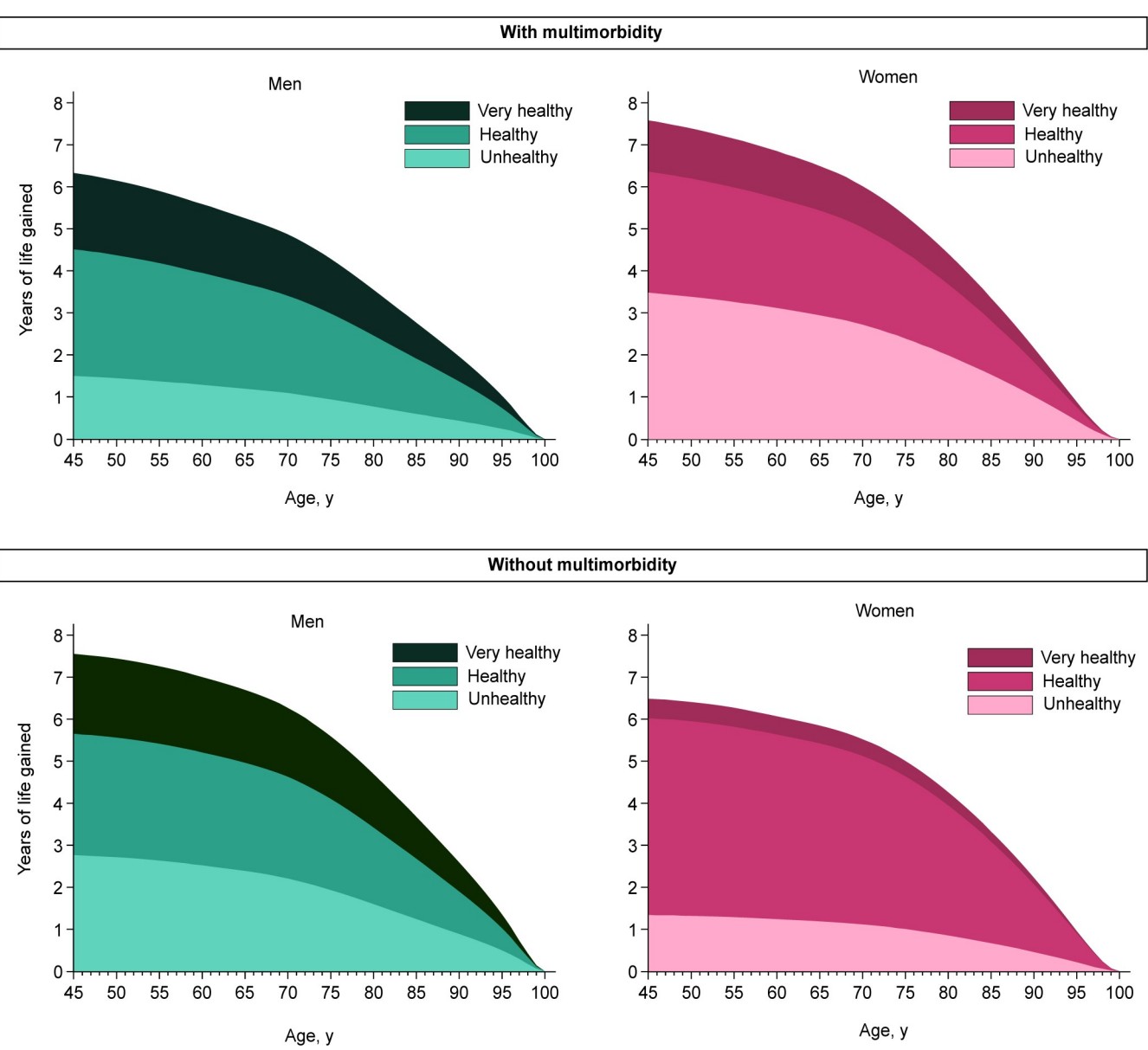

**Fig 2. Years of life gained by lifestyle score.** Reference group is the very unhealthy group. Models adjusted for ethnicity (white, non-white), working status (working, retired, other), deprivation (continuous), body mass index (continuous), sedentary time (continuous).

## Discussion

Our results indicate that in participants with the healthiest lifestyle score, at 45 years the average life expectancy was about 7.6 years longer in men and 6.5 years longer in women compared to those reporting the lowest lifestyle score; conversely, the impact of multimorbidity was approximately 1-year difference: 6.3 years in men and 7.6 in women. These findings have relevant individual, clinical, and public health implications as the results suggest that a healthier lifestyle is similarly associated with longevity regardless of the presence of multimorbidity. Our results also confirmed that not all lifestyle risk factors are equal, and most of the reduction in life expectancy was related to smoking: at 45 years, current smokers had an estimated 5 to 6 years shorter life

**Table 3. Survival by individual lifestyle factor.**

| Healthy lifestyle factor | With multimorbidity | | | | Without multimorbidity | | | |
|---|---|---|---|---|---|---|---|---|
| | Men (*n* = 43,448) | *P* value | Women (*n* = 50,298) | *P* value | Men (*n* = 175,380) | *P* value | Women (*n* = 211,814) | *P* value |
| **Regular physical activity** | | | | | | | | |
| No: No. of deaths/participants | 1,561 / 21,770 | | 1,065 / 30,167 | | 1,910 / 70,847 | | 1,571 / 107,161 | |
| Yes: No. of deaths/participants | 1,058 / 21,678 | | 494 / 20,131 | | 2,083 / 104,533 | | 1,264 / 104,653 | |
| HR (95% CI), Yes vs. No (reference) | 0.73 (0.67 to 0.79) | <0.001 | 0.77 (0.69 to 0.86) | <0.001 | 0.78 (0.73 to 0.83) | <0.001 | 0.87 (0.81 to 0.94) | <0.001 |
| Years of life gained [95% CI], 45 y | 2.49 [1.75 to 3.24] | <0.001 | 1.88 [1.08 to 2.68] | <0.001 | 1.80 [1.30 to 2.31] | <0.001 | 0.88 [0.39 to 1.36] | <0.001 |
| Years of life gained [95% CI], 65 y | 2.11 [1.47 to 2.76] | <0.001 | 1.62 [0.92 to 2.31] | <0.001 | 1.63 [1.17 to 2.08] | <0.001 | 0.79 [0.36 to 1.24] | <0.001 |
| **Smoking** | | | | | | | | |
| Smoker: No. of deaths/participants | 518 / 5,254 | | 269 / 4,859 | | 924 / 21,544 | | 447 / 18,151 | |
| No current smoking: No. of deaths/participants | 2,101 / 38,194 | | 1,290 / 45,439 | | 3,069 / 153,836 | | 2,388 / 193,663 | |
| HR (95% CI), No vs. Yes (reference) | 0.54 (0.49 to 0.60) | <0.001 | 0.48 (0.42 to 0.55) | <0.001 | 0.45 (0.41 to 0.48) | <0.001 | 0.44 (0.40 to 0.49) | <0.001 |
| Years of life gained [95% CI], 45 y | 4.94 [3.83 to 6.06] | <0.001 | 5.94 [4.61 to 7.27] | <0.001 | 5.88 [4.98 to 6.77] | <0.001 | 5.78 [4.83 to 6.72] | <0.001 |
| Years of life gained [95% CI], 65 y | 4.09 [3.13 to 5.04] | <0.001 | 5.09 [3.94 to 6.24] | <0.001 | 5.21 [4.38 to 6.03] | <0.001 | 5.21 [4.34 to 6.07] | <0.001 |
| **Healthy diet** | | | | | | | | |
| No: No. of deaths/participants | 1,789 / 28,903 | | 912 / 28,109 | | 2,867 / 121,358 | | 1,574 / 119,890 | |
| Yes: No. of deaths/participants | 830 / 14,545 | | 647 / 22,189 | | 1,126 / 54,022 | | 1,261 / 91,924 | |
| HR (95% CI), Yes vs. No (reference) | 0.93 (0.85 to 1.01) | 0.072 | 0.91 (0.82 to 1.01) | 0.065 | 0.88 (0.82 to 0.94) | <0.001 | 0.97 (0.90 to 1.05) | 0.494 |
| Years of life gained [95% CI], 45 y | 0.61 [−0.06 to 1.28] | 0.074 | 0.70 [−0.04 to 1.44] | 0.063 | 0.90 [0.39 to 1.40] | 0.001 | 0.17 [−0.31 to 0.64] | 0.493 |
| Years of life gained [95% CI], 65 y | 0.51 [−0.05 to 1.08] | 0.076 | 0.60 [−0.04 to 1.24] | 0.066 | 0.81 [0.35 to 1.26] | 0.001 | 0.15 [−0.28 to 0.58] | 0.504 |
| **Alcohol consumption** | | | | | | | | |
| Excess: No. of deaths/participants | 1,157 / 20,612 | | 281 / 10,812 | | 2,156 / 91,566 | | 704 / 57,434 | |
| None/moderate: No. of deaths/participants | 1,462 / 22,836 | | 1,278 / 39,486 | | 1,837 / 83,814 | | 2,131 / 154,380 | |
| HR (95% CI), None/moderate vs. Excess (reference) | 1.09 (1.01 to 1.18) | 0.029 | 1.15 (1.01 to 1.31) | 0.041 | 0.95 (0.89 to 1.01) | 0.123 | 1.03 (0.94 to 1.12) | 0.560 |
| Years of life gained [95% CI], 45 y | −0.69 [−1.32 to −0.06] | 0.032 | −0.98 [−1.90 to −0.05] | 0.038 | 0.35 [−0.10 to 0.80] | 0.127 | −0.16 [−0.70 to 0.38] | 0.573 |
| Years of life gained [95% CI], 65 y | −0.58 [−1.12 to −0.05] | 0.033 | −0.84 [−1.64 to −0.04] | 0.039 | 0.32 [−0.09 to 0.72] | 0.121 | −0.15 [−0.64 to 0.34] | 0.560 |

Regular physical activity: ≥500 MET-minutes/week; Healthy diet: at least 5 portions of fruit and vegetables every day; None/moderate alcohol consumption: 0 to 14 units of alcohol a week. Model adjusted for ethnicity (white, non-white), working status (working, retired, other), deprivation (continuous), body mass index (continuous), sedentary time (continuous), and all other healthy lifestyle factors. The reference for years of life gained is the same used for HR.

**Abbreviations:** CI, confidence interval; HR, hazard ratio; MET, metabolic equivalent of task; ref, reference

expectancy versus non-current smokers; in comparison, regular physical activity was associated with 1 to 2.5 longer life expectancy versus those not reporting physical activity, while uncertain and smaller associations were observed for healthy diet and alcohol intake.

To our knowledge, this is the first study to quantify whether the risk of death associated with individual and combined risk factors (accounting for their heterogeneous prognostic relevance) was dependent on the presence of multimorbidity. In terms of relative risk, a previous meta-analysis included 15 studies and found that a combination of at least 4 healthy lifestyle factors was associated with a 66% (95% CI 58%–73%) lower risk of mortality [21]. Our results for the healthiest group indicated a 60% lower risk of mortality compared to the unhealthiest group in people with multimorbidity and a 65% lower risk in those without multimorbidity. Moreover, when we used a similar score (count of lifestyle factors), our result indicated a risk reduction ranging from 66% to 71% in relation to sex and presence of multimorbidity, in noticeable agreement with the pooled meta-analytical estimate.

In our systematic search, we found 13 relevant studies, all of which showed a positive association between a healthy lifestyle and life expectancy (S1 Table) [5–7,10–19]. A study in Sweden stratified analyses by the presence of chronic conditions: comparing individuals with low (normal weight, never smoked, participation in at least one leisure activity, and a rich or moderate social network) versus high (overweight or underweight, current or former smokers, no participation in leisure activities, and a limited or poor social network) risk profile, differences in life expectancy were 4.7 years if they had one or more chronic conditions and 3 years if they had no chronic conditions [10]—, though this study population was small ($n$ = 1,661) and included participants over the age of 75 years. Other studies included individuals from the general population and did not investigate differences by multimorbidity status. The results from the general population showed that a combined healthy lifestyle was associated with a longer life expectancy between 5.4 to 18.9 years, compared to the unhealthiest group.

Most of the estimates are higher compared to our study (ranging from 6.3 to 7.6 years), possibly because the definition of a healthy lifestyle was mainly based on non-weighted scores: greater differences comparing healthiest versus unhealthiest groups were indeed observed also in our study when using a non-weighted score. When each risk factor is first dichotomised (score 0: absent; score 1: present) and an overall score obtained as the sum of each score, an equivalent impact of the lifestyle factors on the risk of the outcome is assumed; while this approach has arguably a more immediate public health interpretation, the resulting associations may be larger when participants are grouped into "healthy" (all favourable lifestyle factors) versus "unhealthy" (all unfavourable lifestyle factors). However, it should be also noted that the close agreement between our estimates and those reported in 2 very recent studies (indicating differences in life expectancy between 7.1 and 9.4 years in women and 8.0 and 9.9 years in men comparing healthiest versus unhealthiest using non-weighted sum scores) [18,19] would suggest that, beyond the metric used to define the score, other factors are relevant as well. To our knowledge, only one study from Canada used the individual lifestyle mortality risks when predicting life expectancy [11,12].

The lifestyle factors chosen in this study were smoking and alcohol consumption, physical activity, and nutrition, as these health-related behaviours are related to several individual chronic diseases and are modifiable [3,20]. We found that not smoking had the largest impact on life expectancy for people with and without multimorbidity, similar to studies from the general population [5]. This emphasises the importance of smoking cessation. A healthy diet was defined as eating at least 5 portions of a variety of fruit and vegetables every day [29], as it has been suggested to have beneficial impact on health. A meta-analysis found that a high diet score that included fruit and vegetable intake was associated with a significant reduction in the risk of all-cause mortality, cardiovascular disease, cancer, and type 2 diabetes mellitus [36]. For alcohol intake, we found no meaningful difference in life expectancy: this could be a reflection of participants underreporting alcohol intake. Previous literature reports mixed results about alcohol consumption and risk of death, also quantified in terms of life expectancy [6,7,11].

Multimorbidity is a complex concept. The National Institute for Health and Care Excellence (NICE) UK has recently released guidelines for the assessment and management of people with multimorbidity: the key message from these guidelines is the individualised care [2]. However, whilst a tailored, individual approach mainly focuses on the management of pharmacological interventions, it remains unclear whether and to what extent unhealthy lifestyle behaviours are associated with a higher risk of death in patients with multimorbidity. In this respect, our study significantly contributes to the current evidence: in fact, by providing strong evidence using relative and absolute measures that a healthy lifestyle is equally important in people with and without multimorbidity, it suggests that public health recommendations about engaging in a healthy lifestyle to reduce the risk of developing chronic long-term

conditions equally apply to patients who have already multimorbidity, confirming the importance of a healthy lifestyle throughout the entire lifespan. While multimorbidity is more prevalent in young and middle-aged adults living in the most socioeconomically deprived areas [1], where engaging in a healthy lifestyle could be more difficult, our study also found that certain lifestyle factors are more relevant than others; therefore, public health policies could focus on few, stronger risk factors (i.e., smoking) rather than on costly strategies addressing multiple risk factors. Similarly, when it is proven difficult to reduce all risk factors, individual decision of healthcare professionals may focus on stronger determinants of life expectancy, thus individualising the care of patients with multimorbidity in line with NICE guidance.

This study has several limitations. Firstly, participants from the UK Biobank were volunteers with slightly higher representation from affluent groups; therefore, participants may not be completely representative of the UK population [37]. While the evidence of low generalisability of UK Biobank is documented [38], participants need not be representative of the "target" populations when estimating relative risk factor associations, as expected from a theoretical point of view [39,40] and empirically demonstrated specifically for UK Biobank [41]. Absolute estimates, conversely, are related to the mortality rates in the sample population: as mortality rates in UK Biobank are lower than those in the general population [38] and the relative estimates are applicable to the general population, the differences in years of life quantified in our analyses are likely smaller than those in the general population, further underlining the significant potential benefit of a healthy lifestyle. Second, although participants who died within the first 2 years of follow-up were excluded to reduce the risk of reverse causation [24], it is still possible that participants with multimorbidity may generally be less well, which could result in unhealthy lifestyle behaviours and a higher mortality rate, or adherence to a healthier lifestyle may be associated to a greater adherence to medications. Third, the lifestyle factors were assessed at a single time point, which did not take into account lifestyle changes before or after assessment, and the study was limited to mortality end point. Fourth, lifestyle behaviours are all self-reported measures, which could lead to inaccurate responses, although most large epidemiological studies rely on self-reported questionnaires; however, self-reported physical activity has been found to have a moderate correlation with objective accelerometer measures [42]. Fifth, we did not include other healthy lifestyle factors that could also have an independent association such as sleep duration, other dietary variables (including red or processed meat consumption), or sedentary time. However, in our analyses, we did adjust for sedentary time. Sixth, there is currently no standard definition of multimorbidity [3]. We defined multimorbidity as the presence of 2 or more chronic conditions among 36 conditions that are the core entities in several multimorbidity measures [1,26,27]. Although some studies used a larger number of conditions, we opted for 2 or more as this is the most common approach [3]. Moreover, we searched among the most common QoF diseases to enhance the generalisability of the results. It is also worth noting that, in a previous study, using 2 different methods to define multimorbidity (accounting for the frequency of comorbidities and for self-reported overall health—a proxy of disease severity) showed consistent results regardless of the definition used [8]. Yet we recognise that it remains unclear to what extent the number of conditions or some clusters of multimorbidity modify the association between healthy lifestyles and life expectancy [27]. In the attempt to define a more coherent and homogeneous group of conditions designating multimorbidity, we also explored associations in participants with cardiometabolic multimorbidity. However, we could not consistently compare the results across the 2 definitions because there were very few participants with cardiometabolic multimorbidity: in our analysis, 5.4% had stroke or ischaemic heart disease, whereas in 2017 in the UK, the prevalence ranged from 6.6% in the age group 45–54 years to 21.6% in the age group 65–74 years [43]. Therefore, whether lifestyle factors and an overall healthier lifestyle is differently

associated with life expectancy in relation to the pathophysiological characteristics of the chronic conditions should be explored in further studies. Finally, this was an observational study, and causality cannot be demonstrated.

The overall large sample size, which allowed estimations of the life expectancy by multimorbidity status and sex, is a strength of this study. Another major strength is the reporting of relative as well as absolute measures; absolute measures, particularly how many years of additional life could be gained due to a healthy lifestyle, are easy to interpret and could motivate individuals when considering a lifestyle change. Additionally, we used a weighted healthy lifestyle score as main exposure and complemented the main analysis with sensitivity investigations employing a non-weighted score to assess the robustness of our result, enhance the public health message, and facilitate the comparison with previous literature. Lastly, we based our healthy lifestyle factors on recommended national guidelines for the general population, although personalised lifestyle programs should also consider an individual patient's characteristics and comorbidities [44].

In conclusion, our findings suggest that engaging in a healthy lifestyle could significantly improve life expectancy regardless of the presence of multimorbidity.

## Supporting information

**S1 Text. List of the 36 chronic conditions included within the definition of multimorbidity.**
(DOCX)

**S2 Text. Weighted healthy lifestyle score.**
(DOCX)

**S3 Text. Missing lifestyle and covariate data.**
(DOCX)

**S4 Text. Continuous weighted healthy lifestyle score.**
(DOCX)

**S5 Text. Matching.**
(DOCX)

**S1 Table. Previous studies investigating combined lifestyle factors and life expectancy.**
(DOCX)

**S2 Table. Summary of main and sensitivity analyses.**
(DOCX)

**S3 Table. Most to least prevalent chronic conditions, by sex.**
(DOCX)

**S4 Table. Number of participants by total number of chronic conditions.**
(DOCX)

**S5 Table. Survival using the weighted score obtained from a random one-third of the population.**
(DOCX)

**S6 Table. Survival using the weighted score following imputation of missing data.**
(DOCX)

**S7 Table. Survival using individual lifestyle factor following imputation of missing data.**
(DOCX)

**S8 Table. Survival using the continuous weighted lifestyle score (CIs).** CI, confidence interval
(DOCX)

**S9 Table. Survival using the continuous weighted lifestyle score obtained from a random one-third of the population (CIs).** CI, confidence interval
(DOCX)

**S10 Table. Survival using the continuous weighted lifestyle score following imputing missing data (CIs).** CI, confidence interval
(DOCX)

**S11 Table. Survival using weighted score by cardiometabolic multimorbidity.**
(DOCX)

**S12 Table. Survival using individual lifestyle factor by cardiometabolic multimorbidity.**
(DOCX)

**S13 Table. Survival using number of healthy lifestyle risk factors (score 0–4) by multimorbidity.**
(DOCX)

**S14 Table. Survival using number of healthy lifestyle risk factors (score 0–4) by cardiometabolic multimorbidity.**
(DOCX)

**S15 Table. Survival in the matched cohort.**
(DOCX)

**S1 Fig. Flow chart of participants included in the study.**
(DOCX)

**S2 Fig. Estimated residual life expectancy using the continuous weighted lifestyle score.**
(DOCX)

**S3 Fig. Estimated residual life expectancy using the continuous weighted lifestyle score obtained from a random one-third of the population.**
(DOCX)

**S4 Fig. Estimated residual life expectancy using the continuous weighted lifestyle score following imputation of missing data.**
(DOCX)

**S5 Fig. Years of life gained in the matched cohort.**
(DOCX)

**S1 Checklist. STROBE Checklist.** STROBE, Strengthening the Reporting of Observational Studies in Epidemiology
(DOCX)

## Acknowledgments

This research has been conducted using the UK Biobank Resource (Reference 14614). The views expressed are those of the author(s) and not necessarily those of the National Institute for Health Research (NIHR) or the Department of Health and Social Care.

## Author Contributions

**Conceptualization:** Kamlesh Khunti, Nafeesa N. Dhalwani.

**Data curation:** Nafeesa N. Dhalwani.

**Formal analysis:** Yogini V. Chudasama.

**Investigation:** Yogini V. Chudasama.

**Methodology:** Yogini V. Chudasama, Kamlesh Khunti, Clare L. Gillies, Francesco Zaccardi.

**Project administration:** Yogini V. Chudasama.

**Supervision:** Kamlesh Khunti, Clare L. Gillies, Francesco Zaccardi.

**Writing – original draft:** Yogini V. Chudasama.

**Writing – review & editing:** Yogini V. Chudasama, Kamlesh Khunti, Clare L. Gillies, Nafeesa N. Dhalwani, Melanie J. Davies, Thomas Yates, Francesco Zaccardi.

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
