## [Editor Report · Decision Letter 0]

14 Apr 2020

Dear Dr Chudasama, 

Thank you for submitting your manuscript entitled "Healthy Lifestyle and Life Expectancy in people with Multimorbidity: a UK Biobank prospective study" for consideration by PLOS Medicine.

Your manuscript has now been evaluated by the PLOS Medicine editorial staff as well as by an academic editor with relevant expertise and I am writing to let you know that we would like to send your submission out for external peer review.

Kind regards,

Artur Arikainen

Associate Editor

PLOS Medicine

---

## [Decision Letter · Decision Letter 1]

11 May 2020

Dear Dr. Chudasama,

Thank you very much for submitting your manuscript "Healthy Lifestyle and Life Expectancy in people with Multimorbidity: a UK Biobank prospective study" (PMEDICINE-D-20-01260R1) for consideration at PLOS Medicine. 

[LINK]

In light of these reviews, I am afraid that we will not be able to accept the manuscript for publication in the journal in its current form, but we would like to consider a revised version that addresses the reviewers' and editors' comments. Obviously we cannot make any decision about publication until we have seen the revised manuscript and your response, and we plan to seek re-review by one or more of the reviewers. 

We expect to receive your revised manuscript by Jun 01 2020 11:59PM. Please email us (plosmedicine@plos.org) if you have any questions or concerns.

We look forward to receiving your revised manuscript. 

Sincerely,

Artur Arikainen, 

Associate Editor 

PLOS Medicine

plosmedicine.org

1. We advise you to carefully respond to all of the reviewer comments, as this will be taken into consideration when deciding whether to accept your manuscript for publication. Most importantly, please address the concerns raised by reviewers #1 and #2 regarding the statistical methodology employed in your study.

2. Please remove the word "prospective" from the title (we believe that your paper reports a retrospective analysis of a prospectively gathered dataset).

3. Abstract:

a. Please include the study design, population demographics (eg. age range, sex), and dates during which the study data were collected.

b. In the last sentence of the Abstract Methods and Findings section, please describe the main limitation(s) of the study's methodology.

c. In the Methods and Findings subsection of your abstract, please summarize the factors adjusted for.

d. In the Conclusions subsection of your abstract, please write "... healthier lifestyle was associated ...".

4. Please use the "Vancouver" style for reference formatting, and see our website for other reference guidelines https://journals.plos.org/plosmedicine/s/submission-guidelines#loc-references.

a. Citations in the main text should come before punctuation, e.g., "... multimorbidity measures [1,21,24].

b. In your reference list, please abbreviate journal names consistently (e.g., "PLoS Med.").

5. In the Abstract and throughout the main text, please include p values alongside CIs for your numerical data.

6. Please avoid use of the term “effect” when describing your findings of association.

7. Please remove the data, funding, author contributions, and competing interests statements from page 18 – these are published from corresponding fields on the submission form.

8. In your STROBE checklist, please use section and paragraph numbers, rather than page numbers. Please also add the following statement, or similar, to the Methods: "This study is reported as per the Strengthening the Reporting of Observational Studies in Epidemiology (STROBE) guideline (S1 Checklist)."

9. Please include line numbers throughout your manuscript.

10. We believe you refer to the UK Biobank ethics approval in your methods section. Please also mention the ethics situation for the present study (e.g., cite approval by local IRB). 

11. Early in the Results section, please write "fewer participants". 

----

Comments from the reviewers:

Reviewer #1: I confine my remarks to statistical aspects of this paper. 

There is one major problem that, unfortunately, means that all the analysis has to be redone.

The authors have categorized every continuous variable. This is a mistake. Categorizing continuous variable increases both type I and type II error, it also introduces a kind of magical thinking - i..e. that something amazing happens right at the cutpoint. Frank Harrell, in *Regression Modelling Strategies* listed 11 problems that categorizing independent variables can cause and summed up "nothing could be more disastrous". I wrote a blog post demonstrating some of these problems graphically https://medium.com/@peterflom/what-happens-when-we-categorize-an-independent-variable-in-regression-77d4c5862b6c

All the variables should be left continuous. Splines can be used to look for nonlinearities. 

These changes would affect all of the subsequent write up, so I will wait for a revision to do a review of those parts. 

Peter Flom

Reviewer #2: This paper reports data on lifestyle and life expectancy in multimorbidity from the UK Biobank. This is an important topic and the dataset is sufficiently large to address the study question. I have the following comments for the author to consider:

1. The design is likely to introduce reverse causation bias in particularly because the authors have chosen a very broad definition for multimorbidity (any two or more of the 36 health conditions which vary in terms of severity). For example, a multimorbidity case with 2+ severe and disabling diseases may limited ability to exercise unlike another multimorbidity case with 2 mild health conditions; the association of physical inactivity with life expectancy will be overestimated in this case as the baseline difference in mortality risk between the two cases is not accounted for (ie the participant with 2+ severe conditions has a higher risk of dying independently of physical activity). The authors' attempt to reduce this kind of bias by excluding the first years of follow-up is a good but only partial solution. For this reason, I suggest they run a sensitivity analysis using a more homogeneous definition for multimorbidity - e.g. by looking the associations of lifestyle factors &score with life expectancy in participants with cardiometabolic multimorbidity (ie a combination of cvd and diabetes).

2. The description of multimorbidity definition seems insufficient. It remains unclear how the authors decided which 36 chronic conditions they included in the definition. Why 36 rather than some other quantity and why these specific diseases? Is this a new definition or used also previously? Multimorbidity is a key variable in this paper, so the rationale for the definition should be clear.

3. The authors use a weighted lifestyle score which may introduce circularity bias. To obtain the weights, the authors first compute beta coefficients for each dichotomised lifestyle factor-mortality association. Then they construct a weighted lifestyle score by taking the sum of dichotomised lifestyle factors multiplied by the beta coefficient obtained from the mortality analysis. With this weighted lifestyle score, they estimate differences in life expectancy between those with higher and lower weighted lifestyle score - the main study question. These differences are expected because the weights for the exposure were based on information (ie mortality) from the outcome (life expectancy) - hence the circularity. I suggest that the authors run complementary analyses using a simple sum of dichotomised lifestyle factors as the exposure (range from 0 to 4). Unike the weighted lifestyle score, this indicator will allow comparison of the present findings to those from other studies in the field and it is not subject to circularity.

4. The authors have previously published on physical activity and life expectancy in multimorbidity using UK Biobank (BMC Med 2019) - this study should be noted in the introduction. The same in the description of the assessment of physical activity in this paper - did the authors use the same operationalision? Are the findings on physical activity and life expectancy the same as in the previous paper?

5. Further details are needed on how winsorizing was done as there are many options.

6. How the cut points for 'very unhealthy', 'unhealthy' etc for the lifestyle score categories were chosen?

7. I am surprised by the prevalence of chronic conditions. Why cancer is more common than diabetes in men? Why cancer is more common than depression in women? Are there figures correct; sat least, they seem not to correspond to those observed in the general population. If correct, some discussion is needed on the reliability of measuring diseases using self-reports in the UK Biobank.

8. Discussion, first para. Two main findings are described. However, I do not think the comparison of lifestyle score vs multimorbidity in terms of which is more strongly associated with life expectance is meaningful. With such a broad definition of multimorbidity (any 2+ conditions from a list 36 diseases), the reduction in life expectancy is heavily affected by the specific distribution of the 36 conditions in this highly-selected study population. The finding is by no means generalisable. Thus, I would drop that from the synopsis of the main findings. The other main finding is that "not all lifestyle risk factors are equal" - this has long been known and has been well documented, so I suggest the authors also drop that point. In my opinion, the main finding of this study is that a heathy lifestyle is equally important in term of life expectancy for people with and without multimorbidity. This is a novel and surprising finding which the author should highlight more as it shows how important these factors are for the prognosis/outcome of multimorbidity.

9. Discussion, 2nd para. Here results from a supplementary analysis of unweighted lifestyle score (the sum of lifestyle risk factors) would allow a more direct comparison for other studies.

10. Limitations section. A bit more discussion on generalisability is needed. The 5% response rate in the UK Biobank is exceptionally low by any standards. Selection has been shown to have affected disease prevalence in the cohort. But the key issue here is whether selection is likely to have affected associations between lifestyle and life expectancy. There are studies comparing risk factor-disease outcome associations in UK Biobank and studies with conventional response rates which could help to evaluate this.

11. Several recent studies have examined lifestyle scores in relation to disease-free life expectancy and the results are well in agreement with the current figures on life expectancy - approximately 10+-2 years difference between people with the healthiest versus unhealthiest lifestyle factors (e.g. Zaninotto et al Sci Rep 2020, Nyberg et al JAMA Intern Med, Li et al BMJ 2020). The authors might consider highlighting this close agreement in results across health span and life span.

12. Final paragraph of the discussion, the last sentence. It is well-known that risk factors are not equally strongly associated with life expectancy or mortality - highlighting this as a main conclusion makes this paper look quite non-innovative. I suggest dropping the last sentence.

Reviewer #3: In this study, the authors determined the effect of adherence to a healthy lifestyle on life expectancy in adults with and without known co-morbidities. Using the data from U.K. Biobank, the authors concluded that regardless of the presence of multimorbidity, engaging in a healthier lifestyle is associated with up to 8 years longer life. The study is of great public health importance.

I have the following comments and suggestions: 

1. Given that these results are of great interest to public health, I suggest that scoring of adherence to a healthy lifestyle should be defined differently and simplified. For each lifestyle factor studied, the participants get a score of 1 if they met the healthy definition and 0 if they not. Then, sum these 4 scores and create an overall index of healthy lifestyle ranging from 0-4, with higher scores indicating a healthier lifestyle. The advantage of this simple score is that we get a sense of risk reduction or increased life expectancy if the population shifts to a healthier lifestyle (e.g., from 2 to 4 healthy lifestyle factors). Weighing the score is a sound method that the authors clearly explain in the Discussion, but does not align well with the public health interest.

2. The group with multimorbidity is very heterogeneous, while the group without co-morbidity is homogenous. The study population with multimorbidity included adults with less life-threatening conditions such as glaucoma, hypertension, sinusitis, rheumatoid arthritis, and those with diseases such as heart disease, stroke, cancer, dementia that are leading causes of death. I suggest authors create subgroups of people with multimorbidity and take into account the severity of the disease. Or, authors may apply a weighted score to account for the disease severity.

3. The authors compare people with and without multimorbidity regarding the effect of lifestyle factors and life expectancy, but how different are those groups concerning sample size, demographic, lifestyle, and other clinical factors? To have a fair comparison between groups, the authors should match people with and without multimorbidity and conduct the analysis.

4. The authors report more than 50% of the study population as very healthy according to the weighted score. In the U.S., about 5% of the study population met the overall healthy lifestyle (Li et al. BMJ 2020; 368).

[LINK]

---

## [Decision Letter · Decision Letter 2]

30 Jun 2020

Dear Dr. Chudasama,

Thank you very much for submitting your manuscript "Healthy Lifestyle and Life Expectancy in people with Multimorbidity: a UK Biobank study" (PMEDICINE-D-20-01260R2) for consideration at PLOS Medicine. 

Your paper was evaluated by a senior editor and discussed among all the editors here. It was also discussed with an academic editor with relevant expertise, and sent once more to independent reviewers, including a statistical reviewer. The reviews are appended at the bottom of this email and any accompanying reviewer attachments can be seen via the link below:

[LINK]

In light of these reviews, I am afraid that we will still not be able to accept the manuscript for publication in the journal in its current form, but we would like to consider a revised version that addresses the reviewers' and editors' comments. Obviously we cannot make any decision about publication until we have seen the revised manuscript and your response, and we plan to seek re-review by one or more of the reviewers. 

We expect to receive your revised manuscript by Jul 21 2020 11:59PM. Please email us (plosmedicine@plos.org) if you have any questions or concerns.

We look forward to receiving your revised manuscript. 

Sincerely,

Artur Arikainen, 

Associate Editor 

PLOS Medicine

plosmedicine.org

1. Please respond to the comments of reviewer #3, specifically relating to an additional matching/specificity analysis.

2. Abstract: 

a. Line 53: Given the high p value for this particular result, please rephrase the following to say that the association was not significant: “At 45 years, in men with multimorbidity an unhealthy score was associated with a gain of 1.5 (95% CI: -0.3, 3.3; P=0.572) additional life…”

b. In the last sentence of the Abstract Methods and Findings section, please describe the main limitation(s) of the study's methodology more clearly, and use the word “limitation(s)”.

c. Line 65: Please replace “contributed to” with “correlated with”.

3. Please remove spaces in your citation callouts, eg. “…mental health conditions [1,2],…”

4. Please move the Ethics statement from page 21 to the Methods section.

5. Please move the Data statement from page 21 to the submission form.

6. Please provide more access details (eg. URL, DOI, or issue/page nos.) for references 12, 16, and 34.

------------

Comments from the reviewers:

Reviewer #1: The authors have addressed my concerns and I now recommend publication

Peter Flom

Reviewer #3: I thank the authors for responding to my comments.

I noted that when the analysis was focused on individuals with diabetes, heart disease and stroke, authors reported "imprecise HR and years of life gained estimates" due to the limited number of people with events. Does this indicate a presence of selection bias in the study? The authors are studying life expectancy, but the number of people with life-threatening diseases such as heart disease and stroke does not allow authors to conduct a rigorous analysis among this group? The authors may compare the prevalence of the cardiovascular disease in the UK biobank with national statistics in the UK.

Concluding that the effect of lifestyle factors is similar in people with and without multimorbidity is simplified, in my opinion. First, there is no analysis to compare head-to-head the role of lifestyle factors on life expectancy between the groups with and without multimorbidity. The similarities or differences in life expectancy in each group could be explained by other factors and not necessarily to the adherence to a healthy lifestyle. For this reason, I recommended that authors conduct a matching analysis (as a sensitivity analysis), in which they will select people with multimorbidity with similar lifestyle scores and other characteristics (age, gender, BMI, social status) to people without multimorbidity. As a result of matching, they will create two "identical" groups in terms of sample size, lifestyle, and other characteristics, but different from the presence of multimorbidities. Then, in each group, separately, they will compare the role of adherence to a healthy lifestyle in life expectancy. Second, there is no information about the severity of diseases and the role of medications in people with multimorbidities. It could be that those who adhere to a healthier lifestyle could have less severe conditions and proper adherence to medication, suggesting that these findings could be attributed to disease severity and treatment management, not lifestyle factors adherence. Third, from the public health perspective, the guidelines that authors based on the lifestyle score (e.g., physical activity >150 minutes/week of moderate activity or 75 minutes of vigorous activity) are for primary prevention of chronic diseases such as cardiovascular disease (e.g., heart disease and stroke) and not secondary prevention. For example, according to the American Heart Association/American Stroke Association, physical activity recommendations for stroke survivors should be customized for each individual and should promote low- to moderate-intensity aerobic activity (Stroke. 2014;45:2532-2553).

[LINK]

---

## [Decision Letter · Decision Letter 3]

21 Jul 2020

Dear Dr. Chudasama,

Thank you very much for re-submitting your manuscript "Healthy Lifestyle and Life Expectancy in people with Multimorbidity: a UK Biobank study" (PMEDICINE-D-20-01260R3) for review by PLOS Medicine.

I have discussed the paper with my colleagues and the academic editor and it was also seen again by one reviewer. I am pleased to say that provided the remaining editorial and production issues are dealt with we are planning to accept the paper for publication in the journal.

[LINK]

We look forward to receiving the revised manuscript by Jul 28 2020 11:59PM. 

Sincerely,

Artur Arikainen, 

Associate Editor 

PLOS Medicine

plosmedicine.org

Requests from Editors:

1. Please update the Title to: “Healthy Lifestyle and Life Expectancy in people with Multimorbidity in the UK Biobank: a longitudinal cohort study”

2. Abstract:

a. Line 43: Please use square brackets when nesting inside other brackets: “…(median age of 58 years [range 38-73], 46% male, 95% white)…”

b. At line 55, please adapt the text to "... significantly associated with a gain of 4.5 life years ...".

c. Please add another limitation to the end of the Methods and Findings subsection, eg. participants not being representative of the UK as a whole.

d. At line 64, please begin the sentence with "In this analysis of data from the UK Biobank, we found that ..." or similar.

e. Rather than "up to 8 years" at line 65, we suggest quoting the observed maxima for men and women (i.e., 6.3 and 7.6 years).

3. Methods: Please cite the prospective protocol on line 146 for clarity.

4. Tables 2, 3: Please include p values for all relevant results.

5. Please upload the STROBE S1 Checklist as a separate file.

----

Comments from Reviewers:

Reviewer #3: Thank you for addressing my comments. I have no additional comments or suggestions.

[LINK]

---

## [Editor Report · Decision Letter 4]

18 Aug 2020

Dear Dr Chudasama, 

On behalf of my colleagues and the academic editor, Dr. Sanjay Basu, I am delighted to inform you that your manuscript entitled "Healthy Lifestyle and Life Expectancy in people with Multimorbidity in the UK Biobank: a longitudinal cohort study" (PMEDICINE-D-20-01260R4) has been accepted for publication in PLOS Medicine. 

PRODUCTION PROCESS

PRESS

PROFILE INFORMATION

Thank you again for submitting the manuscript to PLOS Medicine. We look forward to publishing it. 

Best wishes, 

Artur Arikainen, 

Associate Editor 

PLOS Medicine

plosmedicine.org